# The Evolution Characteristics of Systemic Risk in China’s Stock Market Based on a Dynamic Complex Network

**DOI:** 10.3390/e22060614

**Published:** 2020-06-02

**Authors:** Yong Shi, Yuanchun Zheng, Kun Guo, Zhenni Jin, Zili Huang

**Affiliations:** 1School of Computer Science and Technology, University of Chinese Academy of Sciences, Beijing 100190, China; yshi@ucas.ac.cn (Y.S.); zhengyuanchun14@mails.ucas.ac.cn (Y.Z.); 2Key Laboratory of Big Data Mining and Knowledge Management, Chinese Academy of Sciences, Beijing 100190, China; jinzhenni18@mails.ucas.ac.cn; 3Research Center on Fictitious Economy & Data Science, Chinese Academy of Sciences, Beijing 100190, China; 4College of Information Science and Technology, University of Nebraska at Omaha, Omaha, NE 68182, USA; 5School of Economics and Management, University of Chinese Academy of Sciences, Beijing 100190, China; 6Sino-Danish College, University of Chinese Academy of Sciences, Beijing 100049, China; 7Geisel School of Medicine, Dartmouth College, Hanover, NH 03755, USA; zili.huang.gr@dartmouth.edu

**Keywords:** complex network, systemic risk, structural entropy, stock market, EMD

## Abstract

The stock market is a complex system with unpredictable stock price fluctuations. When the positive feedback in the market amplifies, the systemic risk will increase rapidly. During the last 30 years of development, the mechanism and governance system of China’s stock market have been constantly improving, but irrational shocks have still appeared suddenly in the last decade, making investment decisions risky. Therefore, based on the daily return of all a-shares in China, this paper constructs a dynamic complex network of individual stocks, and represents the systemic risk of the market using the average weighting degree, as well as the adjusted structural entropy, of the network. In order to eliminate the influence of disturbance factors, empirical mode decomposition (EMD) and grey relational analysis (GRA) are used to decompose and reconstruct the sequences to obtain the evolution trend and periodic fluctuation of systemic risk. The results show that the systemic risk of China’s stock market as a whole shows a downward trend, and the periodic fluctuation of systemic risk has a long-term equilibrium relationship with the abnormal fluctuation of the stock market. Further, each rise of systemic risk corresponds to external factor shocks and internal structural problems.

## 1. Introduction

The stock market is a typical complex system with multiple stock prices fluctuating from equilibrium to deviation and to equilibrium again. A large number of heterogeneous investors buy and sell stocks frequently, making the relationships between different stocks unpredictable. In most scenarios, owing to some factors like herd effect, investors’ investment strategies converge [1,2]; when some investors buy a stock, other investors tend to buy the same one, and furthermore, when the vast majority of investors buy or sell stocks, other investors usually follow this action. At the same time, different listed companies are another heterogeneous agent in the stock market. On the one hand, the economic exchanges between listed companies will lead to the linkage of their stock prices. On the other hand, similar actions by investors on similar stocks can cause herd behavior between different stock prices. When the prices of a large number of stocks in the market tend to be consistent, it means that the herd effect in the market is higher, and the stock market is more likely to fluctuate excessively and consistently, leading to higher market systemic risks [3,4].

In former studies, the capital asset pricing model (CAPM) framework was usually used to analyze financial systematic risks as a basic theory [5,6,7,8]. According to CAPM, risks can be divided into systematic risk (or market risk) and non-systematic risk, while the latter can be diminished through investment portfolios. The systematic risk often refers to pervasive, far-reaching, perpetual market risk, which can be measured by the variance of the portfolio (Beta) altogether. Therefore, most studies on systematic risk are based on Beta values. Although this theory is widely adopted, it usually comes with a number of hypotheses, such as homogenous investors in capital markets. However, in modern financial markets, different investors generally have different degrees of rationality, ability to obtain information, and sensitivity to prices, that is, investors are usually heterogenous. Hence, CAPM may not be a reasonable model in the real complex world [9,10]. More importantly, this paper focuses on the systemic risk, which reflects the stability of the system and the characteristics of risk transmission among individuals in a certain complex system.

A complex network, which is based on physics and mathematics theory, can tackle complicated practical problems [11]. It is especially suitable for modeling, analysis, and calculation in complex finance systems [12]. Nowadays, the literature on applying complex networks to finance is growing in size, and complex networks have become important tools in the finance field [13]. After 30 years of development, China’s stock market is growing in scale and vitality, while the market operation mechanism and management system are constantly improving. Nevertheless, there have been several typical bear and bull markets in recent years, and systemic risk in the stock market has risen periodically. Therefore, a dynamic complex network of individual stocks in China’s stock market is constructed in this paper to measure the dynamic systemic risk of China’s stock market. Then, the tendency evolution and cycle change characteristics of systemic risk are explored. 

The structure of this paper is as follows. Section 2 summarizes the applications of complex networks in the field of economy and finance; Section 3 introduces the data and methodology used in this paper; Section 4 proposes the empirical results and analysis; and the conclusions and some discussion are given in Section 5.

## 2. Related Works

Construction of the network consists of two important steps, defining nodes and defining edges. In previous studies, nodes are usually represented by different agents in the financial market, that is, stocks or bonds, and edges are symbolized by the relationship between such agents. Pearson’s correlation coefficient is the most common and easiest way to measure the correlation between two entities in the financial market [14,15,16,17,18,19,20]. For example, McDonald et al. used Pearson correlation coefficient to construct a currency-related network in the global foreign exchange market and obtained temporary dominant or dependent currency information [16]. In addition, other correlation coefficients, such as Spearman rank-order correlation coefficient [21], multifractal detrended cross-correlation analysis (MFCCA) [22,23], multifractal detrended fluctuation analysis (MFDCA) [24], and cophenetic correlation coefficient (CCC) [25], have also been put forward. Furthermore, correlation can also be defined by some econometric methods, such as the Granger causality test [26,27], cointegration test [28], dynamic correlation coefficient with GARCH (DCC-GARCH) [29], and so on.

After the definition of edges, some filter methods for choosing the important edges should be applied. Otherwise, the complex network will be very large and complicated, which is not conducive to subsequent analysis. Minimum spanning tree (MST) can be used for this purpose. After MST operation, the complex network will retain only *N* − 1 edges, where *N* is the number of the nodes, which greatly facilitates the study of the network topology. At present, MST is most commonly used to simplify the financial complex network [14,15,18,19,20,22,29,30,31]. For example, in 1999, Mantegna first proposed that MST could be used to search for important edges in the stock market network, and a stock market topology with economic significance could be obtained [14]. Except for MST, other greedy algorithms similar to MST, such as planar maximally filtered graph (PMFG) [28], can also be used as filter methods. Furthermore, setting thresholds and only retaining edges with a correlation coefficient greater than the threshold can work as a filter too [17].

After construction of the financial network with the above methods, some structural characteristic information can be obtained by analyzing the network indicators. Previous scholars have found that stock markets in different countries have similar topologies. Zhuang et al. used stock data from the Shanghai stock market from 2002 to 2004 to build an undirected and unweighted network. It was found that China’s stock networks had the typical statistical features of complex networks, that is, small world and scale-free property [17]. Tu also selected Chinese stock data and built a stock market complex network. The author calculated some network indicators like degree centrality, PageRank, hyperlink induced topic search (HITS), local clustering coefficient, K-shell, and so on, finding the driving forces of China’s financial market [28]. Chi et al. selected the U.S. stock market data to build several weighted networks (using price, price returns, and trading volumes as edges separately), finding that the U.S. stock price networks are scale-free networks, which means the variation of stock prices is strongly influenced by a relatively small number of stocks [29]. Caraiani used the complex network to study the returns of major European emerging countries’ stock markets, which also had the characteristic of being scale-free [24]. Furthermore, complex network studies were also conducted on the Japanese stock market [32], Korean stock market [33], Hong Kong stock market [34], and so on.

On the basis of the topological structure of the financial network, the systemic risk in the financial market can be measured and the impact of crisis on the financial system can be analyzed. In general, crisis will change the topology of the financial network, and a more complex topology (or a larger value of structural entropy) often comes with a greater systemic risk [35]. Onnela et al., for example, built a U.S. stock market network with MST and conducted a tree structure analysis on it, finding that the financial crisis caused the tree length to shrink, which means the topological structure of the stock market experienced a downfall when facing systemic risk [15]. Long et al. selected CSI 300 data in China, established a dynamically correlated stock industry network with MST, and studied its connection characteristics and topological structure. The results reveal that industries with large betweenness centrality, closeness centrality, clustering coefficient, and small node occupancy layer are associated with greater systemic risk contribution [29]. He and Deem found that the systemic risk and recession will lead to a more hierarchical structure of global trade networks [25]. Bardoscia et al. also found that when the connectedness in bank networks increases, the financial system’s ability to deal with systemic risk will deteriorate [36].

To sum up, according to various distance definitions and filter methods, different financial complex networks can be constructed, thus obtaining different measures of systemic risk. Pearson’s correlation coefficients are simple and intuitive, and can effectively measure the correlation between different stocks in the Chinese stock market. Since its establishment in 1990, China’s stock market has been rapidly developing for nearly 30 years. While the internal operating mechanism adjusts constantly, China’s stock market is also subject to external impacts, so the systemic risk of the market is dynamic. This paper constructs a dynamic complex network based on the stocks daily return data; symbolizes the systemic risk of China’s stock market by complex network indicators; decomposes and reconstructs the sequence into three components including trend, cycle, and high frequency disturbance; and, finally, examines the evolution of China’s stock market systemic risk.

## 3. Data and Methodology

This paper establishes a dynamic complex network model utilizing the daily data of all a-shares in Shanghai and Shenzhen stock markets. Because of the small number of stocks at the beginning of the establishment of the Chinese stock market, we selected 90 trading days before the first trading day in 1997 as the starting point, and 29 February 2020 as the end point.

The methodology is shown in Figure 1. First, the correlation coefficients between return rates of stocks were used as edge weights to build dynamic complex networks with a window period of 90 days and 1 day for step. Then, the average weight and structural entropy of the network in each day can be obtained. The ratio of the stocks with weight in the top 10% to the average weight in each window period can also be calculated and defined as the concentration ratio of important stock. Therefore, four network indexes could be derived. Next, these four network indexes were combined with the stock market index and 0–1 standardization before empirical mode decomposition (EMD) was performed. Through the above process, the original sequences were divided into a number of intrinsic mode functions (IMFs). Then, the results were reconstructed with grey relational analysis (GRA), making each sequence have three items, that is, tendency, cycle, and disturbance. Finally, the statistical analysis of the three components was conducted in order to explore the development of China’s stock market and the evolution characteristics of systemic risk. Modeling of the complex network, EMD, and GRA is introduced as follows.

### 3.1. Construction of the Complex Network

A complex network consists of several nodes and edges linking them. The node is the basic element of a complex network, which is the abstract expression of an “individual” in the real world. The edge is an expression of the relationship between the elements and can be given weight according to the extent of the relationships. Here, $w_{ij}$ represents the weight of the edge linking node i and node j, where i, j=1,2,3,…,n and n is the number of nodes in a certain network. For an undirected network,
(1)$$w_{ij}=w_{ji}$$

We can also use the weighted degree to represent the importance of nodes, which is defined as
(2)$$dw_{i}=\sum_{j\in v\left(i\right)}w_{ij}$$
where $v_{\left(i\right)}$ is the set of nodes linking to node i. The larger the weighted degree, the stronger the degree of correlation with other nodes and the more important the node.

We use the return rates of a-share stocks on China’s stock market as the network nodes and construct the network using correlation coefficient $\rho_{ij}$ as the edge weight.
(3)$$w_{ij}=\rho_{ij}=<X_{it}\cdot X_{jt}>$$

Here, $\left\{X_{it},\;i=1,2,\cdots ,n;t=1,2,\cdots ,T\right\}$ is the original stock return rates data and <⋯> indicates a time-average over the T data points for each time series.

After we get $w_{ij}$, we calculate the average weight, top 10 nodes weight, and concentration ratio below:(4)$$average\;weight=\frac{1}{n}\overset{n}{\underset{i=1}{\sum }}dw_{i}$$
(5)$$top\;10\;nodes=\frac{1}{10}\underset{i\in top\left(i\right)}{\sum }dw_{i}$$
(6)$$concentration\;ratio=\frac{top\;10\;nodes}{average\;weight}$$
where $top_{\left(i\right)}$ means the nodes i with the top 10 weights ($dw_{i}$).

Furthermore, we calculate the network’s structural entropy, which is often used to measure the complexity of the complex network system [37]. However, as the structural entropy of the all-connected network is constant, it is meaningless for our analysis, so we need to remove the edge of weak correlation to get a non-all-connected network for calculating the structural entropy.

The threshold value of the correlation coefficient is set at 0.4. If the absolute value of the correlation coefficient $\rho_{ij}$, that is, $w_{ij}$, is less than 0.4, this edge will be cut off, and we will get a non-fully connected network to calculate the structural entropy $E_{deg}$ under each window [37]:(7)$$E_{deg}=-k\overset{N}{\underset{i=1}{\sum }}p_{i}\log p_{i}$$
where N is the total number of nodes in the network; k is Boltzmann’s constant; and $p_{i}$ can be calculated by the number of edges connecting to node i, namely, the degree of node i:(8)$$p_{i}=\frac{degree\left(i\right)}{\sum_{i=1}^{N}degree\left(i\right)}$$

### 3.2. Empirical Mode Decomposition

Combining the three network indexes with China’s stock market index gives four input data, named as $\left\{Y_{kt},\;k=1,2,3,4;t=1,2,\cdots ,T\right\}$. $Y_{kt}$ have to be 0–1 standardized, owing to significant differences at the numerical level, that is,
(9)$$Z_{kt}=\frac{Y_{kt}-{\bar{Y}}_{k}}{std\left(Y_{k}\right)}$$

For the signal Z(t), the upper and lower envelopes are determined by local maximum and minimum values of the cubic spline interpolation. $m_{1}$ is the mean of envelopes. Subtracting $m_{1}$ from Z(t) yields a new sequence $h_{1}$. If $h_{1}$ is steady (does not have a negative local maximum or positive local minimum), it is denoted as the intrinsic mode function ($imf_{1}$). If $h_{1}$ is not steady, it is decomposed again, until steady series is attained, which is denoted as $imf_{1}$. Then, $m_{1}$ replaces the original Z(t) and $m_{2}$ is the mean of the envelopes of $m_{1}$, and $m_{1}$ is similarly decomposed. Repeating these processes K times gives $imf_{k}$, that is,
(10)$$imf_{k}=imf_{\left(k-1\right)}-m_{k}$$

Finally, let res denote the residual of Z(t) and all imfs:(11)$$res=X\left(t\right)-imf_{1}-imf_{2}-\cdots -imf_{K}$$
where imfs and res could be extracted for the GRA process.

### 3.3. Grey Relational Analysis

The grey relational analysis was first put forward by Deng J L in 1989 [38]. His grey relational degree model, which is usually called the grey relative correlation degree, mainly focused on the influence of distance between points in the system.

The grey relative correlation degree formula is given by Equation (12).
(12)$$r_{ij}^{1}=\frac{1}{N}\overset{N}{\underset{t=1}{\sum }}\frac{\begin{array}{c} min\\ j \end{array}\begin{array}{c} min\\ t \end{array}\left|d_{i}\left(t\right)-d_{j}\left(t\right)\right|+\rho \begin{array}{c} max\\ j \end{array}\begin{array}{c} max\\ t \end{array}\left|d_{i}\left(t\right)-d_{j}\left(t\right)\right|}{\left|d_{i}\left(t\right)-d_{j}\left(t\right)\right|+\rho \begin{array}{c} max\\ j \end{array}\begin{array}{c} max\\ t \end{array}\left|d_{i}\left(t\right)-d_{j}\left(t\right)\right|}$$
where $d_{i}\left(t\right)$ is the reference series; $d_{j}\left(t\right)$ is the compared series; and ρ is the distinguishing coefficient, which is usually equal to 0.5.

In order to overcome the weakness of the grey relative correlation degree, the absolute correlation degree was proposed by Mei (1992) [39]. The formula is given by Equation (13).
(13)$$r_{ij}^{2}=\frac{1}{N-1}\overset{N-1}{\underset{t=1}{\sum }}\frac{1}{1+\left|d_{i}\left(t+1\right)-d_{i}\left(t\right)+d_{j}\left(t+1\right)-d_{j}\left(t\right)\right|}$$

Considering the weakness and strength, we used the grey comprehensive relational degree to classify the noise terms and market fluctuation terms. The formula of the grey comprehensive relational degree is given by Equation (14):(14)$$r_{ij}=\beta r_{ij}^{1}+(1-\beta )r_{ij}^{2}$$
where β is the weight of the grey relative relational degree, which is valued as 0.5.

## 4. Empirical Analysis

### 4.1. Dynamic Characteristics of Complex Networks

Figure 2a compares the three average weight related indicator of the dynamic complex network with the dynamic evolution of the Shanghai composite index standardized by setting it as 1000 on the first trading day of 1997. It can be seen that the average weight of the complex network and the average weight of the top 10 stocks have strong synchronization, with a high correlation of 0.9896. Therefore, both of them can be used as proxy indicators of systemic risk. However, the concentration ratio is not consistent with the overall systemic risk. The concentration of risk is relatively low when the systemic risk is high, which means the risk is relatively decentralized. Furthermore, the concentration ratio and the average weight are significantly negatively correlated with a correlation coefficient of −0.91329. In this way, we will focus on using the index of the average weight to measure the systemic risk of the Chinese stock market.

It can also be seen from Figure 2a that, although there is a correlation between two average weight indexes (all and top 10) and the stock index, the coefficients, −0.1370 and −0.0829, are relatively small. This proved that the level of systemic risk is not determined by the move of overall price trend.

In order to further investigate the relationship between the systemic risk represented by the average weight, the Beta value (β) obtained by the CAPM model, and the stock average variance (V), we estimated $\beta_{t}$ and $V_{t}$ as follows:(15)$$X_{kt}=r_{f}+\beta_{kt}(Y_{t}-r_{f})+e_{kt}$$
(16)$$\beta_{t}=\frac{1}{N}\overset{N}{\underset{k=1}{\sum }}\beta_{kt}$$
(17)$$V_{t}=\frac{1}{N}\overset{N}{\underset{k=1}{\sum }}\left[\frac{1}{T}\overset{T}{\underset{m=1}{\sum }}{\left(X_{km}-\bar{X_{kt}}\right)}^{2}\right]$$
where N is the total number of stocks; T is the length of the sliding window; $r_{f}$ is the risk-free interest rate, which was set to 3%; $X_{kt}$ is the return of the kth stock in the sliding window t; $Y_{t}$ is the return of the stock index, which is symbolized for market return and is represented by 000001.SH; $\beta_{kt}$ is calculated by MLS with $Y_{t}$ and $X_{kt}$; $e_{kt}$ is the error term; $\beta_{t}$ is the average of all individual stocks’ Beta; and $V_{t}$ is the average variance of all stocks in sliding window t.

In Figure 2b, we compare the systemic risk with Beta and stock variance, finding that these three have different moving trends, which shows that our systemic risk index can catch unique market fluctuations. Furthermore, the systemic risk index was ahead of Beta in several stages, such as from June 2006 to July 2008 or from July 2015 to August 2017, which shows that our systemic risk index has a certain risk pre-warning ability.

We further compared the systemic risk represented by average weight with the volatility index (VIX) of China and the U.S. stock market. Considering the Chinese VIX cannot cover the above research range, the U.S. VIX was selected for comparison purposes. The correlation coefficient between the two VIX in this range is significantly positive, but the coefficient is only 0.5626.

Figure 3a presents the great differences in the trend of VIX between China and the United States. It can be seen that the correlation coefficient between average weight and Chinese VIX is 0.4763 during the interval since the Chinese VIX launched. It is noteworthy that the volatility index leads the systemic risk index to a certain extent. This is confirmed by the results obtained from the cross-correlation analysis with the maximum coefficient of 0.7469, corresponding to lags of 55 days (which means current systemic risk is highly related to the VIX from 55 days prior). However, this is mainly because the systemic risk index constructed in this paper was compiled using the sliding window method, with the window length of 90 days, so the systemic risk index of a certain time, *t* actually represents the systemic risk of the previous 90 days. 

In fact, the complex network characteristics of individual stocks are effective at reflecting the systemic risk of the market. To verify this, we calculated the 90-day averages for VIX, which are shown in Figure 3b. It can be seen that the systemic risk index constructed in this paper is consistent with the 90-day average trend for China’s VIX, and the systemic risk is ahead of China’s VIX after 2017 and is more sensitive, which proves the effectiveness of the systemic risk index derived from the complex network.

Figure 4 shows the comparison between the structural entropy and the number of nodes in a complex network. It can be seen that the structural entropy is highly correlated with the number of nodes, and the correlation coefficient reaches 0.9302. In other words, the increase in system complexity of China’s stock market is mainly caused by the increase in the number of listed companies. Nevertheless, we can also find that, in addition to the overall upward trend, structural entropy also has periodic fluctuations. Therefore, multi-scale analysis is required to determine whether the system complexity represented by structural entropy is related to systemic risk.

### 4.2. Decomposition and Reconstructione

Figure 5 presents the EMD results of the standardized systemic risk index, structural entropy, and stock price index, respectively. It can be seen that the two original sequences are divided into seven IMFs and one residual term, among which the residual term can represent the overall trend of indexes’ evolution to a certain extent, while the IMF of lower frequency can describe the periodic fluctuation of indexes in different time scales, and the IMF of highest frequency represents the stochastic perturbation.

Through EMD, it can be found that the residual term, also known as the trend term, decomposed by structural entropy, represents the growth in the number of network nodes, and the correlation coefficient between this residual term and the number of network nodes can be further improved to 0.9428. When removing the trend term from the original sequence and comparing it to the systemic risk series represented by the average weight, as shown in Figure 6, the highly consistent fluctuations between the two series can be seen, and the correlation coefficient of the two reaches 0.7572. Therefore, adjusted structural entropy, that is, removing the trend term of the network size, can also measure the systemic risk. Nevertheless, owing to the high correlation between these two series, the following analysis only focuses on the systemic risk represented by average weight.

In order to further observe the systemic risk evolution of the Chinese stock market, several IMFs and residual terms obtained from EMD decomposition were combined using the method of grey correlation degree. Figure 7 and Figure 8 present the trend term, cycle term, and random term of systemic risk (average weight) and the stock price index. Then, we focused on the overall trend change and cycle fluctuation of systemic risk in China’s stock market.

For the long-term tendency, we found that the overall trend of the stock price rose steadily, while the systemic risk has been declining slowly throughout the evolution of the Chinese stock market since 1997. This means that, although there is still phased systemic risk in the Chinese stock market, the overall level of systemic risk is declining as the operating mechanism and related regulations are constantly improving.

For the cycle fluctuation, the rise of systemic risk is usually caused by the joint action of external shocks and internal operations, which is manifested in the excessive rise and fall in the stock market. Therefore, the cyclical characteristics of systemic risk have no direct relationship with the fluctuations of the stock market. Thus, we converted the cycle fluctuation of the stock market into the difference from the price mean using (18).
(18)$$cycle\_abs\_stock=abs\left[cycle_{stock}-average\left(cycle_{stock}\right)\right]$$

Considering that *cycle_abs_stock* and *cycle_risk* are both non-stationary, we calculated their first-order differences. The results of Augmented Dickey–Fuller (ADF) tests show that both variables are an integrated of order one. Therefore, cointegration tests can be proposed on the original sequences. The results of Johnson Trace tests show that there are at least two cointegration relationships between the two variables, which confirms that there is a long-term equilibrium relationship between stock price volatility and systemic risk. The equilibrium equation is
(19)cyclel_abs_stock=0.0660cycle_risk+0.0625.

All the coefficients are significant at the 5% significance level, so the volatility of the stock market is positively related to systemic risk from the perspective of long-term equilibrium, which means that, while the stock price deviates from the theoretical value of equilibrium, the systemic risk will be at a high level.

In Figure 9, when the blue line is above 0, the systemic risk is large, while when the blue line is below 0, the systemic risk is small. The red line represents the absolute value of stock price movements, and the red line is clearly ahead of the above-zero parts of the blue line.

Figure 9 shows the cycle evolution of systemic risk, the lead-lag relationship between systemic risk and stock volatility is dynamic owing to the sliding window processing. From the perspective of the whole cycle evolution, we found that there were several periods of high systemic risk in the Chinese stock market since 1997, as described below.

1997–1998: The stock market was in a shock stage during this period. On the one hand, the Chinese stock market was impacted by external factors such as the Asian financial crisis; on the other hand, the operating mechanism at that time was not perfect enough, with frequent insider trading and market manipulation. The systemic risk was at a high level during this period and, therefore, the stock market began to comprehensively reform its trading mechanism in 1998. Although the market fluctuation was not violent from the current perspective, it actually contained many factors causing systemic risk.

2001–2002: The stock market was in a declining bear stage during this period. Owing to the poor performance of high-tech companies, resulting from the burst of the global Internet bubble and the launch of the policy reducing the state-owned shares holding of listed companies, the stock market had a big crash in China. Related departments issued a series of favorable strategies such as reducing interest rates and trading commissions; however, the imperfection of the market led to a number of “black markets”, which brought a high systemic risk.

2007–2008: This period includes both excessive rise and fall of the market. The reform of non-tradable shares in 2005, together with a series of positive policies such as the entry of insurance funds and the appreciation of renminbi (RMB), promoted the rise of the Chinese stock market. However, a lot of speculation by inexperienced individual investors caused a more and more serious herding effect, and the systemic risk was maintained at a high level for a long time. Followed by the global financial crisis brought by the U.S. subprime crisis, with the launch of stock index futures, the Chinese stock market began to reverse to a bear stage, and the systemic risk in this stage also remained at a high level.

2011–2012: This stage was another volatile bear market; the fluctuation of stock price was much smaller than that of the previous stage, but the systemic risk still remained at a similar level. Even though the Chinese economy maintained a high growth rate during this period, the stock market was influenced by the global financial markets, as well as the European debt crisis. The low volatility of the stock market still contained large systemic risks, which were reinforced by the frequent occurrence of black swan events such as a rear-end collision of bullet trains, clenbuterol, and so on.

2015–2016: The market price was rising rapidly in 2015 and the systemic risk was also in a climbing stage. However, a high level of risk still appeared in 2016, which was a stage of rapid and frequent fluctuations. The issuing scale of new stocks increased significantly, driving frequent market shocks such as thousands of shares rising or falling together, two triggering circuit breaker events in a day, and so on. Thus, the overall capital presents a large-scale net outflow, and the investor sentiment fluctuates abnormally.

To summarize, the systemic risk of the stock market will significantly increase in the irrational stages of rise, fall, and frequent shocks. However, extremely high systemic risk is more likely in the cases of collapse and frequent shocks.

## 5. Discussion

Complex networks have been widely used in the field of socio-economic analysis. Most of them focus on the risk contagion of banks and international economic or trade exchanges; however, studies on the stock market are limited. In fact, a complex network provides an important tool for the study of the stock market, which is a self-organizing complex system with multi-agent interactions. The average weight of the complex network can be used to measure the aggregation of positive feedback in the market, so as to measure the overall systemic risk.

On the basis of the data of all a-shares in China, this paper constructs a dynamic complex network of stock correlation, and the change of average weight as well as adjusted structural entropy of the network are used to measure the evolution of systemic risk in China’s stock market. Although, owing to the use of a sliding window, the average weight or structural entropy in fact presents the average systemic risk level in the past 90 days, it also reflects the evolution of systemic risk in China’s stock market for more than 20 years as a whole. The results show that the systemic risk of China’s stock market shows a downward trend on the whole, which is closely related to the continuous improvement of the management system and operation mechanism of the financial market. In addition, there is a long-term equilibrium relationship between the cycle fluctuation of systemic risk and the excessive fluctuation of the stock market. Since 1997, the stages with high systemic risk have appeared with excessive increases, excessive falls, and frequent fluctuations of the stock market. Meanwhile, it can also be seen from Figure 1 that the global stock market began to fluctuate significantly under the influence of the novel coronavirus pneumonia. The Chinese stock market is relatively stable at present, but the systemic risk has been climbing rapidly since the beginning of February. Therefore, we must be alert to the further expansion of the systemic risk of the Chinese stock market under the double impact of internal and external factors.

## Figures and Tables

**Figure 1 entropy-22-00614-f001:**
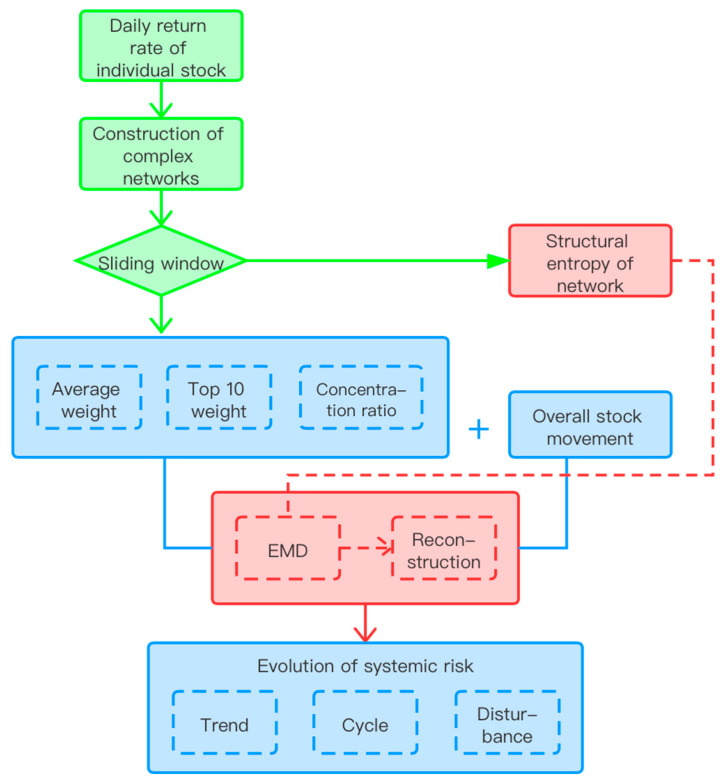
Methodology. EMD, empirical mode decomposition.

**Figure 2 entropy-22-00614-f002:**
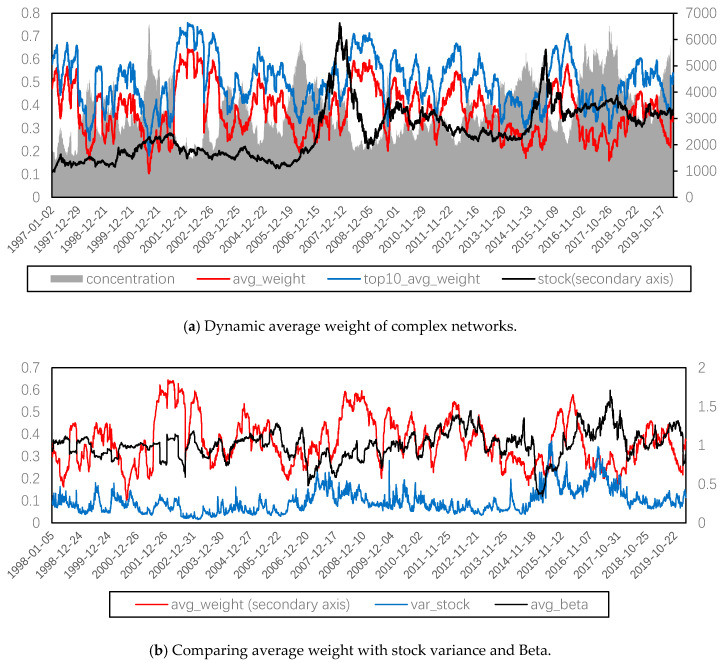
Comparison of complexity measures of systemic risk with other measures.

**Figure 3 entropy-22-00614-f003:**
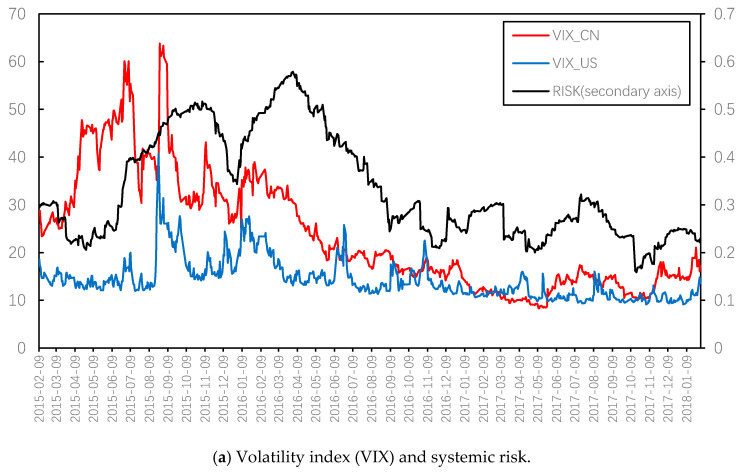
Comparison of systemic risk with VIX.

**Figure 4 entropy-22-00614-f004:**
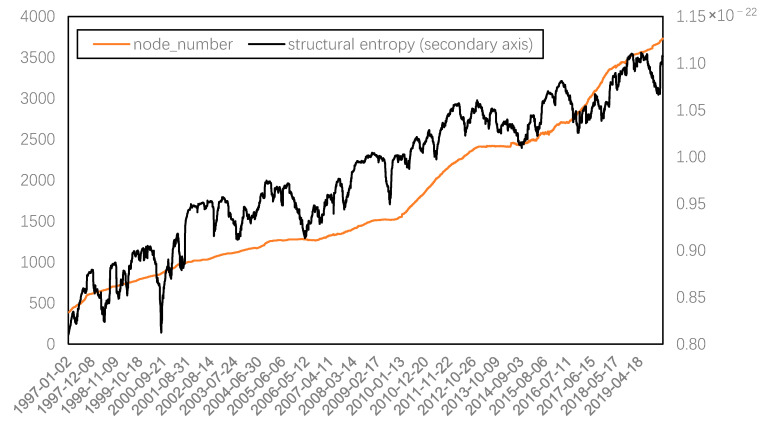
Dynamic structural entropy of complex networks.

**Figure 5 entropy-22-00614-f005:**
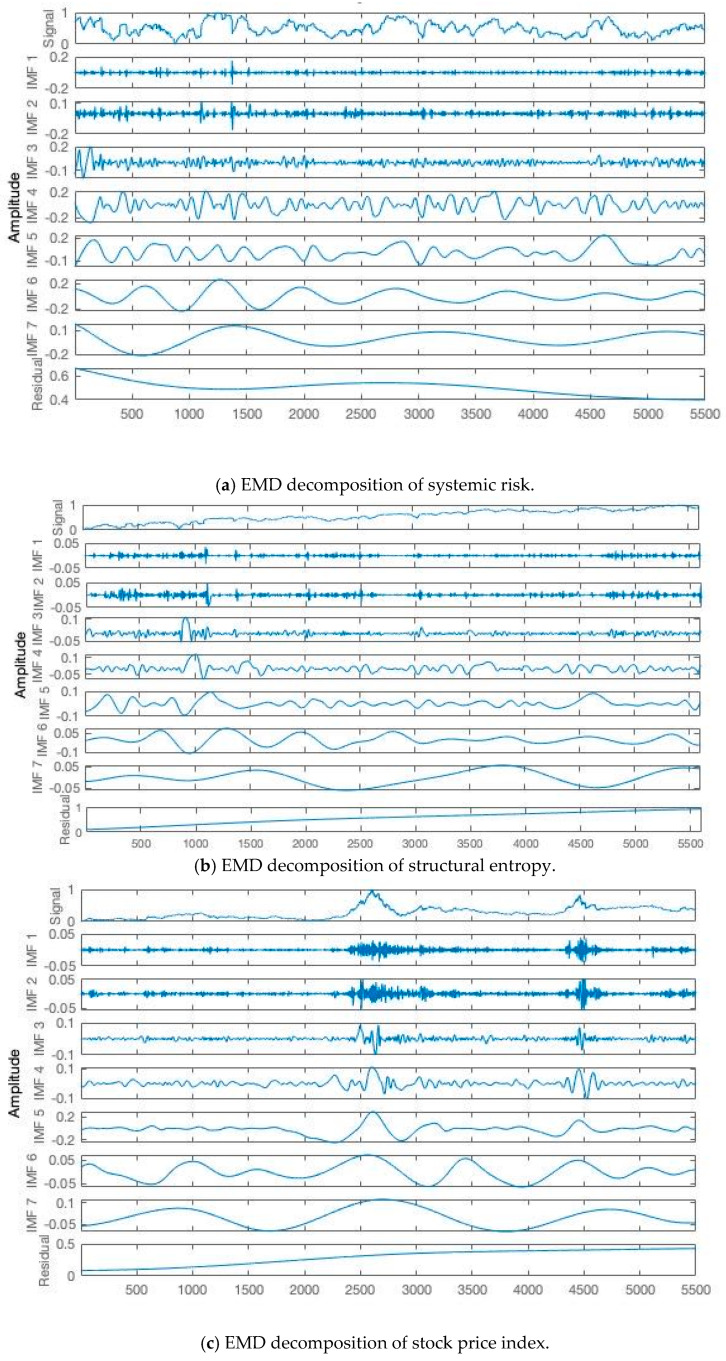
Results of EMD decomposition. IMF, intrinsic mode function.

**Figure 6 entropy-22-00614-f006:**
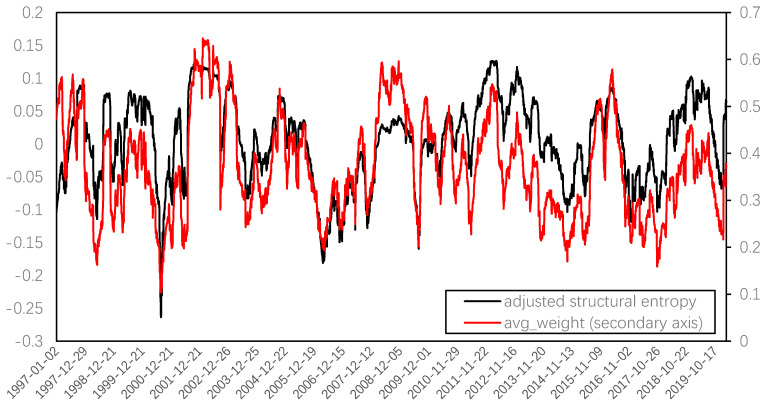
Adjusted structure entropy and average weight.

**Figure 7 entropy-22-00614-f007:**
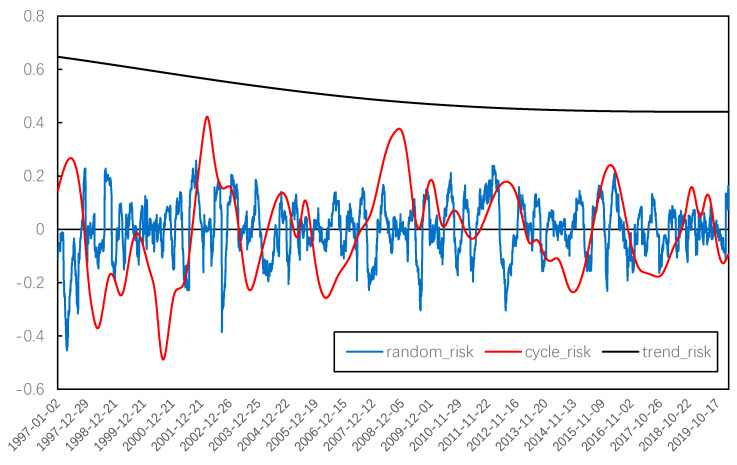
IMFs reconstruction of systemic risk.

**Figure 8 entropy-22-00614-f008:**
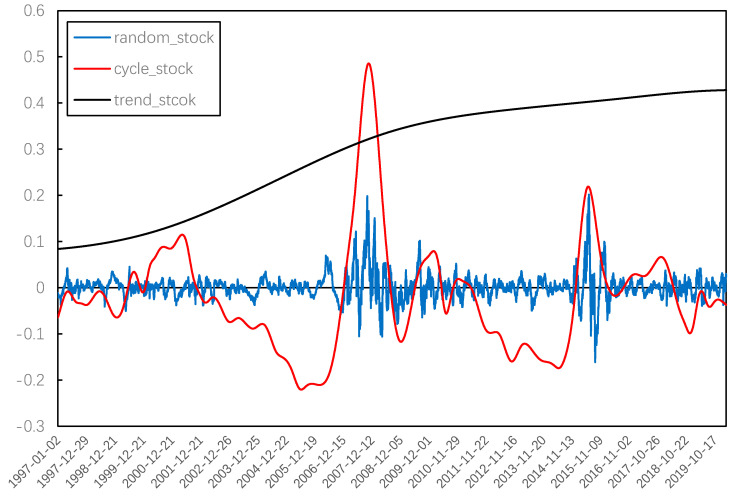
IMFs reconstruction of stock price index.

**Figure 9 entropy-22-00614-f009:**
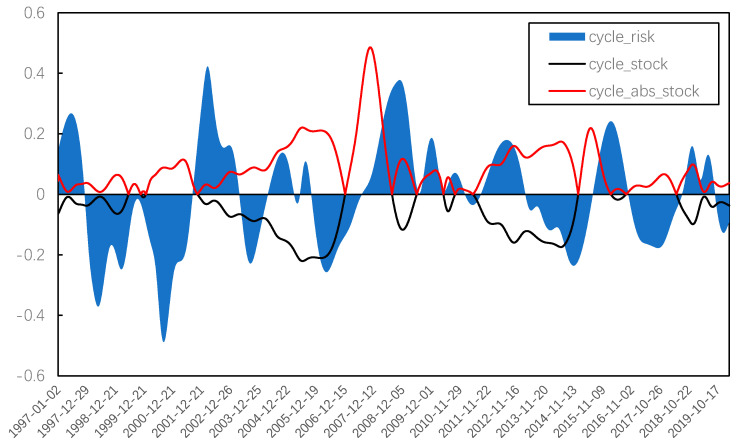
Cycle evolution of systemic risk.

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
