# Peer review of "The Evolution Characteristics of Systemic Risk in China’s Stock Market Based on a Dynamic Complex Network"

_entropy, 2020, doi:10.3390/e22060614_

Round 1
Reviewer 1 Report
The article is very valuable. I rate it highly considering its originality.
My doubts
and objections, raises the method of explaining the problem and presenting the assumptions for research. I wonder how many people involved in investing and analyzing capital markets have the knowledge to fully understand the authors' research?
I would like to ask for fuller explanations related to the importance of research and the presentation of the assumptions for the study, the results obtained and the importance for science.
Based on the article, you can't repeat the authors' research? How to verify test results? This is the weaker part of the article.
Author Response
Thank you very much for pointing out the problems in our manuscript. We have revised it according to your recommendations. We would like to know if there are still somewhere need to be amended.
Point 1: “The article is very valuable. I rate it highly considering its originality. My doubts and objections, raises the method of explaining the problem and presenting the assumptions for research. I wonder how many people involved in investing and analyzing capital markets have the knowledge to fully understand the authors' research? I would like to ask for fuller explanations related to the importance of research and the presentation of the assumptions for the study, the results obtained and the importance for science.”
Response 1:
Thank you very much again for your valuable advice and your recognition of our work. We apologize for the confusion caused by our writing problem. We have major revised the parts of 1. Introduction, 2. Related Works and 4.1. Dynamic characteristics of complex networks in order to make our research easier for readers to understand. The following is a brief introduction of the article's research structure, and we hope that would be helpful for readers repeating our results.
- Research Purpose: To study the systemic risk in a typical complex system, the stock market, we selected the returns of all a-share stocks in Chinese stock market as data and researched the correlations between various stocks to measure how easily risk can be transmitted between stocks.
- Research Object: The focus of this paper is systemic risk, which reflects the stability of the system and the characteristics of risk transmission among individuals in a certain complex system. Note that “systemic risk” and “systematic risk” have some difference, the systematic risk often refers to pervasive, far-reaching, perpetual market risk, which can be measured by the variance of the portfolio, while the systemic risk addressed the risk transmission.
- Research Assumptions and Methods: Based on the assumption that different subjects in the complex system are not homogeneous, the traditional economic methods are not applicable in the complex system, so this paper used the complex network mothed to conduct the research. For example, in most financial papers, capital asset pricing model (CAPM) is often used to study systematic risk. However, in modern financial market, different investors generally have different degrees of rationality, ability to obtain information and sensitivity to prices. So CAPM may not be reasonable in real complex world, and that’s why we applied complexity measures here.
- Modeling Steps: Firstly, we constructed the complex network of stocks with the sliding window method. To illustrate, we set different stocks in the sliding window as nodes, used the correlation between stocks as the weight of the edges, and set the threshold value to retain the strongly correlated edges, so as to complete the network construction. Then, according to the constructed network, different network indicators can be obtained, including average weight, average weight of top 10 nodes, concentration ratio, and structural entropy of network. Then we used these indicators to measure systemic risk, and compared with the other methods that measure systemic risk, such as the VIX, Beta and Variance of individual stocks. It’s found that the network indicators are more sensitive and are moving ahead of the other indicators, which verified the effectiveness of the network indicators. Finally, we use the EMD and GRA to reconstruct the complex network indicators, getting the trend term and period term, so as to study the dynamic evolution characteristics of systemic risk in China's stock market.
- Major Conclusions
- Compared with VIX and Beta, the complexity measures of systemic risk have a certain risk pre-warning ability.
- The systemic risk of China’s stock market as a whole shows a downward trend.
- The periodic fluctuation of systemic risk has a long-term equilibrium relationship with the abnormal fluctuation of stock market.
Point 2: Based on the article, you can't repeat the authors' research? How to verify test results? This is the weaker part of the article.
Response 2:
Thank you for your valuable comments. Our data is all open source and valid, and if readers hope to repeat our work, it is our pleasure to provide our research data. In addition, to verify our results of complexity measures of systemic risk, we compared them with some traditional measures of risk, such as VIX, Beta and stock variance, finding that the complexity measures have a certain risk pre-warning ability.

Reviewer 2 Report
Comment 1.
On line 44, the authors state:
"Based 44 on framework of Capital Asset Pricing Model (CAPM) corporation risk can be divided into different 45 components, including financial leverage, operating risk and others [5]."
The CAPM is a single factor model. A corporation's risk is divided into market risk (i.e., systematic risk) and idiosyncratic risk.
In addition, the authors need to emphasize the fact that the CAPM assumes homogenous investors while this article investigates a complex system with heterogenous investors. The homogeneity assumption can be one reason that the CAPM doesn't work for the real world as the authors mentioned on line 51. This note can guide readers to the motivation of this study smoothly.
Comment 2.
The authors should distinguish terms like systematic risk and systemic risk. Finance literature uses the terms with different meaning. In Fig1, the authors used the term systemic risk rather than systematic risk. (Also on lines 259 and 277, 282. In my opinion, the term systemic risk is closer to what the authors intended to deliver because the proxies suggested by the authors gauge the complexity (entropy) of a system.
Comment 3.
The reason why the stock index has a low correlation with the complexity measure is that stock index (or price) doesn't related with systematic risk. The variance of index returns should be checked. Therefore, VIX can be a good measure for systematic risk.
However, VIX is an option-implied volatility, which measures an expectation about future (forward-looking measure), while the proxy used in this study is calculated by past data. That's why the correlation is low. To be comparable, the authors need to compute systematic risk using the same period data. For example, run the regression r(i) = a + b*IndexRetrun + e(i) for each stock with 90-day window data, and then compute sum of squared regressors (or the variance of b*IndexReturn), which is a systematic risk of individual stock. Finally, the average of them can be a comparable measure of systematic risk level for this study.
Comment 4.
Basically, this study regards each stock as a random element in a complex system (stock market). So, nodes are individual stocks and edges are correlations among them, and roughly speaking, the complexity is defined by average correlations.
The first sentence in the introduction may not suit for this modeling. The authors state:
"Stock market is a typical complex system with multi-agent interaction."
which, sounds like investors are nodes and their trading behaviors are edges. The existence of heterogenous agents can be what makes stock markets complex, but the sentence may make readers misunderstand that this study models individual investors as an element of the system.
Minor comments:
on line 89 Li et al. should be numbered.
on line 142, IMF is not defined before (probably intrinsic mode function).
on line 146, GCA should be spelled.
in eq(8) degree(i) is not defined.
VIX is the name of the S&P option implied volatility index. The term volatility index would be more proper for.
Need a grammar check.
"Complex networks, based on physics and mathematics theory, can tackled complicated practical 52 problems [15]."
Author Response
Thank you very much for pointing out the problems in our manuscript. We have revised it according to your recommendations. We would like to know if there are still somewhere need to be amended.
Point 1: “On line 44, the authors state: "Based 44 on framework of Capital Asset Pricing Model (CAPM) corporation risk can be divided into different 45 components, including financial leverage, operating risk and others [5]." The CAPM is a single factor model. A corporation's risk is divided into market risk (i.e., systematic risk) and idiosyncratic risk. In addition, the authors need to emphasize the fact that the CAPM assumes homogenous investors while this article investigates a complex system with heterogenous investors. The homogeneity assumption can be one reason that the CAPM doesn't work for the real world as the authors mentioned on line 51. This note can guide readers to the motivation of this study smoothly.”
Response 1:
Thank you very much again for your constructive suggestions. We have revised accordingly on Line 48 and Line 53:
“According to CAPM, risks can be divided into systematic risk (or market risk) and non-systematic risk, while the latter can be diminished through investment portfolios.”
“However, in modern financial markets, different investors generally have different degrees of rationality, ability to obtain information and sensitivity to prices, that is, investors are usually heterogenous. Hence, CAPM may not be a reasonable model in the real complex world [9,10].”
Point 2: The authors should distinguish terms like systematic risk and systemic risk. Finance literature uses the terms with different meaning. In Fig1, the authors used the term systemic risk rather than systematic risk. (Also on lines 259 and 277, 282. In my opinion, the term systemic risk is closer to what the authors intended to deliver because the proxies suggested by the authors gauge the complexity (entropy) of a system.
Response 2:
Thank you for your professional advice. Systemic risk is indeed closer to the main idea of the article, so we replaced systematic risk with systemic risk in all other areas except CAPM parts. In addition, on Line 50 and Line 56, we added the definition between systematic risk and systemic risk respectively.
“The systematic risk often refers to pervasive, far-reaching, perpetual market risk.”
“More importantly, this paper focuses on the systemic risk, which reflects the stability of the system and the characteristics of risk transmission among individuals in a certain complex system.”
Point 3: The reason why the stock index has a low correlation with the complexity measure is that stock index (or price) doesn't related with systematic risk. The variance of index returns should be checked. Therefore, VIX can be a good measure for systematic risk. However, VIX is an option-implied volatility, which measures an expectation about future (forward-looking measure), while the proxy used in this study is calculated by past data. That's why the correlation is low. To be comparable, the authors need to compute systematic risk using the same period data. For example, run the regression r(i) = a + b*IndexRetrun + e(i) for each stock with 90-day window data, and then compute sum of squared regressors (or the variance of b*IndexReturn), which is a systematic risk of individual stock. Finally, the average of them can be a comparable measure of systematic risk level for this study.
Response 3:
Thank you for your professional advice. In figure 2b, according to your comments, we tested the relationships between complexity measure of systemic risk and the variance of index returns in eq (17), as well as the Beta value estimated by CAPM in eq(15)-eq(16), finding that these three indicators moving differently. And our systemic risk index is ahead of Beta in several stages, which shows the complexity measure of systemic risk have a certain risk pre-warning ability.
Point 4: Basically, this study regards each stock as a random element in a complex system (stock market). So, nodes are individual stocks and edges are correlations among them, and roughly speaking, the complexity is defined by average correlations. The first sentence in the introduction may not suit for this modeling. The authors state: "Stock market is a typical complex system with multi-agent interaction." which, sounds like investors are nodes and their trading behaviors are edges. The existence of heterogenous agents can be what makes stock markets complex, but the sentence may make readers misunderstand that this study models individual investors as an element of the system.
Response 4:
Thank you for your constructive advice. To avoid reader’s confusion, the original sentence, "Stock market is a typical complex system with multi-agent interaction." has been replaced by "The stock market is a typical complex system with multiple stock prices fluctuating from equilibrium to deviation and to equilibrium again."
In addition, the original sentence, “A large number of heterogeneous investors buy and sell stocks frequently, which can be regarded as multiple interactive games, thus generating various complex scenarios" has been replaced by "A large number of heterogeneous investors buy and sell stocks frequently, making the relationships between different stocks unpredictable."
Minor comments:
- on line 89 Li et al. should be numbered.
- on line 142, IMF is not defined before (probably intrinsic mode function).
- on line 146, GCA should be spelled.
- in eq(8) degree(i) is not defined.
- VIX is the name of the S&P option implied volatility index. The term volatility index would be more proper for.
- Need a grammar check. "Complex networks, based on physics and mathematics theory, can tackled complicated practical 52 problems [15]."
Response:
Thank you for your constructive advice. We have modified the corresponding part as follows:
- Due to Li et al has weak relationship with our paper, we have removed this reference.
- On line 149, IMF have been defined as intrinsic mode function.
- “GCA” has been corrected to “GRA”.
- In eq(8), degree has been defined as the number of edges connecting to node .
- On line 59, the original sentence has been corrected to “A complex network, which is based on physics and mathematics theory, can tackle complicated practical problems.”

Round 2
Reviewer 2 Report
The authors have polished their manuscript, reflecting my comments.
It seems that the quality of the current version can satisfy the criteria for the publication.
This manuscript is a resubmission of an earlier submission. The following is a list of the peer review reports and author responses from that submission.